Exploring the impact of visual stimuli on taste expectations and the role of pull-tab position in food packaging

Shibuya Kenichi shibuya@kansai-u.ac.jp 1 2 3 4
Miyamoto Mana 2 5
1 Faculty of Health and Well-being, Kansai University , Sakai , Japan
2 Department of Health and Nutrition, Niigata University of Health and Welfare , Niigata , Japan
3 Graduate School of Health and Welfare, Niigata University of Health and Welfare , Niigata , Japan
4 Graduate School of Human and Well-being, Kansai University , Sakai , Japan
5 Department of Food Science and Nutrition, Nara Women’s University , Nara , Japan
Antoun Jumana
Electronic publication date: 2025 Dec 4
Publication date: 2025
Volume: 13
Electronic Location ID: e20417
Received 2025 Apr 25; Accepted 2025 Oct 28
Copyright: ©2025 Shibuya and Miyamoto
Copyright year: 2025
Copyright holder: Shibuya and Miyamoto
License: This is an open access article distributed under the terms of the Creative Commons Attribution License, which permits unrestricted use, distribution, reproduction and adaptation in any medium and for any purpose provided that it is properly attributed. For attribution, the original author(s), title, publication source (PeerJ) and either DOI or URL of the article must be cited.
License URL: https://creativecommons.org/licenses/by/4.0/

Keywords: Preference, Graspability, Food, Ease-of-reaching, Canned

Funding: The authors received no funding for this work.

==============================
Background

Preference for an image is often associated with the perceived ease of use of the depicted object.

Aim

This study investigated the impact of visual stimuli on taste expectations, focusing on the presentation of food in images. Specifically, it explored how the presence of a pull-tab—perceived by participants as a tool that makes the product easy to use or open—influences these expectations.

Methods

A sample of 42 right-handed university students (30 women, 12 men; mean age = 20.4 years) evaluated perceived taste expectations of food presented under different conditions using a 9-point Likert scale. The conditions included images of canned food (“Can condition”) with varying pull-tab positions (0°, 90°, 180°, 270°, and Down), as well as images of actual food (“Pineapple condition”). Each participant was shown 12 images per trial in a randomized order and rated their taste expectations immediately after viewing each image. Statistical analyses were conducted using linear mixed models and ANOVA using the Kenward-Roger method.

Results

Taste expectations were significantly lower in the Can condition compared with the Pineapple condition (mean difference = 3.14, SE = 0.23, p < 0.001). Within the Can condition, pull-tab positions had a significant effect on taste expectation ratings (F = 116.94, p < 0.001). The 0° position (pull-tab facing the viewer) received the highest rating (M = 4.87, SD = 1.86), while the 180° position (pull-tab facing away) received the lowest (M = 3.95, SD = 1.73).

Conclusion

These findings underscore the complex relationship between visual presentation and taste expectations, providing valuable insights for optimizing product presentation in marketing strategies.

Introduction

Images of food have been utilized in various research fields, including experimental psychology (cf. Motoki et al., 2018; Shibuya et al., 2020; Shibuya et al., 2022; Spence, Motoki & Petit, 2022). Insights from these studies hold considerable potential for promoting health within the field of public health. When individuals are presented with an edible item even visually the brain promptly evaluates its anticipated taste (e.g., Motoki, Saito & Onuma, 2021; Shibuya et al., 2022; Spence, Motoki & Petit, 2022). Neuroimaging research has demonstrated that viewing food images activates brain regions associated with taste perception and reward processing, such as the insula and the orbitofrontal cortex (Petit, Javornik & Velasco, 2022; Van der Laan et al., 2011). These findings suggest that visual food cues can elicit anticipatory responses related to taste and consumption.

However, the brain’s ability to imagine the taste of food may be influenced by the type of image presented (Piqueras-Fiszman & Spence, 2012). For example, although a food item placed on a plate and the same food enclosed within a container are physically identical, the former may evoke a more vivid mental representation of taste (Piqueras-Fiszman & Spence, 2012). This discrepancy may arise because food presented in a container is often perceived as less appetizing; the brain must initiate a sequence of preparatory actions, such as opening the container, removing the food, grasping it, and bringing it to the mouth. Consequently, food in a container may be perceived as less fresh than food presented on the plate. Nevertheless, it remains unclear whether such perceptual differences persist when comparing canned food with fresh fruit.

Conversely, preference for an image is often influenced by how easily the depicted object appears to be used (Palmer, Gardner & Wickens, 2008). For instance, individuals tend to favor images of tools such as scissors or pots when the functional or manipulable part is oriented toward them (Palmer, Gardner & Wickens, 2008). This suggests that perceived ease of use, even from a visual standpoint, can enhance preference. Similarly, Shibuya et al. (2022) demonstrated that foods that appear easier to grasp are also rated as tastier. In the case of canned food images, a can with the pull-tab facing the viewer may be perceived as easier to open, thereby increasing visual preference. However, previous research has suggested that, when interacting with packaged food, the brain may need to simulate a series of preparatory actions—such as recalling the contents or imagining the opening process—which could interfere with its ability to vividly simulate the expected taste (Palmer, Gardner & Wickens, 2008; Okamoto & Dan, 2007; Spence & Wang, 2015). If visual manipulability indeed shapes preference judgments, then even subtle variations in the orientation of packaging components—such as the pull-tab—may meaningfully influence consumer impressions.

This study hypothesized that (1) images of canned food would receive lower taste-expectation ratings compared with images of uncanned (fresh) food, and (2) among canned food images, those with pull-tabs oriented toward the viewer would receive higher ratings due to the perceived ease of opening. Accordingly, when individuals evaluate the expected taste of food, the orientation of the pull-tab—as perceived from the viewer’s perspective—is predicted to influence judgments through its implied manipulability. Specifically, images in which the pull-tab appears easy to grasp and open are expected to elicit higher taste ratings. This suggests that the ease with which the tactile experience of handling food can be mentally simulated may exert a stronger influence on taste expectations than the perceived mechanical ease of opening itself.

Several potential sources of bias may have influenced the results of this study. One possible bias concerns participants’ perceptions of can-opening difficulty based on the orientation of the pull-tab. For instance, the perceived ease of opening may have shaped participants’ expectations and, in turn, affected their taste ratings. To mitigate this concern, a control condition with a neutral pull-tab orientation was included to isolate the specific effect of pull-tab positioning from other confounding factors. Another potential bias involves participants’ general attitudes toward canned food, as a negative predisposition could have contributed to lower ratings for canned food images compared with those of fresh fruit. This factor was therefore taken into account when interpreting the results. Accordingly, participants’ overall attitudes toward canned food were assessed, and the extent to which these attitudes influenced their taste evaluations was examined.

Materials and Methods

Participants

This study involved 42 right-handed university student volunteers (30 women, 12 men). Handedness was assessed using the Edinburgh Handedness Inventory (Oldfield, 1971). The sample size was determined through a pre-simulation using the simr (Green & MacLeod, 2016) and sjstats (Lüdecke, 2022) packages in R. Drawing on prior studies with similar designs (e.g., Shibuya et al., 2022), we anticipated a medium effect size (f = 0.25) for the primary comparisons. Assuming an alpha level of 0.05 and desired power of 0.80, the analysis indicated that a minimum of 36 participants would be required for a repeated-measures ANOVA with six conditions (corresponding to the six image types). To account for potential attrition and to enhance statistical power, 42 participants were recruited. A post hoc power analysis confirmed that this sample size provided a power of 0.87 to detect medium-sized effects. Only right-handed participants were included to control for potential confounds related to handedness in visual perception and motor imagery. Previous research has shown that left- and right-handed individuals may differ in visual processing and mental rotation abilities, which could influence how object orientations are perceived and evaluated.

None of the participants reported any medical conditions relevant to the study, such as visual agnosia or eating disorders. Participants were aged between 18 and 22 years (mean age = 20.4 years), and all had normal or corrected-to-normal vision. Exclusion criteria included color blindness, uncorrected visual impairments, diagnosed eating disorders, and neurological conditions that could affect visual perception or decision-making. These conditions were assessed using a self-report questionnaire administered prior to the decision-making task. Approval for this study was obtained from the Ethics Committee of Niigata University of Health and Welfare (approval number: 18914-221101). Each participant received a detailed explanation of the study procedures and provided written informed consent before participation.

Protocols

Upon arrival at the laboratory, each participant was directed to a small room and asked to sit in front of a computer for at least 15 min. A unique identification (ID) number was assigned to each participant to ensure anonymity and facilitate data management. Participants were then given a detailed explanation of the experimental tasks and completed them using a computer program developed with PsychoPy (Peirce et al., 2019). During the task, the upper half of the screen displayed images of canned pineapple (“Can condition”) and fresh pineapple (“Pineapple condition”). Figure 1 illustrates the five distinct images used in the experiment: five canned pineapple images with varying pull-tab orientations—(1) 0° (pull-tab facing the viewer), (2) 90° (pull-tab oriented to the left), (3) 180°, (4) 270°, and (5) Down (pull-tab not visible). All canned pineapple images were identical except for the pull-tab orientation and minor processing adjustments made to ensure uniformity. The fresh pineapple image used in the “Pineapple condition” was derived from the canned pineapple image to maintain visual consistency.

Figure 1 Illustrations depicting various orientations of the presented images in the current study.

(A) 0 degrees orientation, where the pull-tab faces towards the viewer (0 deg), (B) 90 degrees orientation with the pull-tab oriented to the left from the viewer (“90 deg”), (C) each subsequent 90-degree rotation leading to the 180-degree position (“180 deg”), (D) 270 degrees orientation (“270 deg”), (E) denoted as “Down,” characterized by the pull tab being not visible, and (F) denoted as the “Pineapple” condition.

The randomization of image presentation was performed individually for each participant using a computer-generated random sequence. This procedure ensured that each participant received a unique and unpredictable order of stimuli, thereby minimizing potential order effects. The randomization was implemented using Python’s random module, with a seed generated from the system clock to ensure true randomness across participants. Each trial consisted of 12 images presented in a randomized order unique to each participant. In this study, “graspability” refers to the perceived ease with which an object can be grasped and manipulated, based solely on visual information. “Ease of use” is defined as the perceived simplicity and efficiency with which an object can be utilized for its intended purpose. Both constructs were measured indirectly through participants’ taste expectation ratings in response to the visual stimuli. The term images specifically refers to the visual stimuli presented to participants, including both canned and fresh pineapple images. The term stimuli encompasses these images, which were used influence participants’ taste evaluations.

Prior to the experiment, participants received detailed instructions on how to use the 9-point Likert scale (Lavin & Lawless, 1998). They were informed that one represented ‘Dislike Extremely,’ five represented ‘Neither Like nor Dislike,’ and nine represented ‘Like Extremely.’ Participants were encouraged to use the full range of the scale and to base their ratings on immediate taste expectations prompted by the visual appearance of the food images. Responses were entered using the keyboard, and all ratings were recorded automatically by the computer program for subsequent analysis. The stimuli were presented on a 27-inch monitor (VG272, Acer, Saitama, Japan) with a resolution of 1,920  × 1,080 pixels and a refresh rate of 240 Hz. A computer (MacBook Pro, Apple, Cupertino, CA, USA) controlled both the presentation of stimuli and data collection. Visual stimuli included a fixation point and images of food, presented at a viewing distance of 40 cm. The luminance of the fixation point was 91.0 cd/m2. Each trial consisted of a 1,000 ms fixation cross display, followed by a 2,000 ms food image presentation, both centered on the computer screen. Participants were then allowed up to 5,000 ms to provide their rating after the image disappeared. This relative brief response window as designed to maintain participants’ concentration and elicit their immediate taste expectations without prolonged deliberation. The inter-trial interval was set at 1,000 ms, resulting in a total trial duration of approximately 9,000 ms.

Statistical analysis

All statistical analyses were performed using R (version 4.1.3; R Core Team, 2022) and the lmerTest package (Kuznetsova, Brockhoff & Christensen, 2017). Data were presented as mean ± standard deviation (SD). To account for data nesting and potential violations of sphericity, and to minimize the risk of Type I errors, a linear mixed-effects model (LMM) was used to examine differences in the evaluation of food images based on the 9-point Likert scale across conditions with hierarchical data (Murayama et al., 2014). The dependent variable, overall liking of the pineapple images, was modeled as a function of the fixed effect Condition (Can and Pineapple) with random intercepts for participant ID. Additionally, a random slope for the effect of Conditions within each image was included. Model selection was conducted using likelihood ratio tests, with the best-fitting model chosen based on the Akaike Information Criterion (AIC) and Bayesian Information Criterion (BIC). The final model exhibited the lowest AIC and BIC, and the likelihood ratio test indicated a significant improvement over simpler models. An analysis of variance (ANOVA) using the Kenward-Roger method was then performed to assess the overall significance of the fixed effect Conditions. Subsequently, within the Pineapple condition, we examined the fixed effect of Images on overall liking using ANOVA with the Kenward-Roger method. Post-hoc pairwise comparisons between images were conducted using the Bonferroni correction to control for multiple comparisons. Statistical significance was set at p < 0.05.

Results

A total of 42 participants (30 females and 12 males; mean age = 20.4 years) were included in the final analysis. Our analysis revealed two primary findings: (1) images of canned food received significantly lower taste expectation ratings than images of fresh food, and (2) within the canned food images, the position of the pull-tab significantly influenced taste expectation ratings. The mean overall liking ratings were 4.35 (SD = 1.77) for the Can condition, and 7.06 (SD = 1.44) for the Pineapple condition. Within the Can condition, the mean ratings for each stimulus were 4.87 (SD = 1.86) for 0°, 4.23 (SD = 1.66) for 90°, 3.95 (SD = 1.73) for 180°, 4.33 (SD = 1.73) for 270°, and 4.38 (SD = 1.77) for the Down position.

Figure 2 presents the overall liking ratings for the two conditions (Can and Pineapple conditions). The fixed effects analysis for Conditions indicated a significant effect (p < 0.001), showing that participants in the Can condition gave significantly lower ratings than those in the Pineapple condition. Furthermore, the ANOVA confirmed the significance of the Conditions factor (p < 0.001), reinforcing its crucial role in explaining the variance of the dependent variable.

Figure 2 Difference in overall liking between Can and Pineapple conditions.

Overall liking was significantly higher in the Pineapple condition than in the Can condition. Statistically significant differences are indicated by asterisks (*; p < 0.05). Error bars represent standard deviation (S.D.).

Figure 3 illustrates the overall liking ratings for the six images (0°, 90°, 180°, 270°, Down and Pineapple). A significant difference in liking was found among the six images (p < 0.001). The results of the post-hoc tests are presented in Table 1. Post-hoc analyses revealed significant differences between most pull-tab positions (Table 1). The 0° position (pull-tab facing the viewer) received the highest rating (M = 4.87, SD = 1.86), which was significantly higher than all other positions (all ps < 0.001). The 180° position (pull-tab facing away) received the lowest rating (M = 3.95, SD = 1.73), significantly lower than all other positions except the 90° position (p = 0.089). Interestingly, the Down position (M = 4.38, SD = 1.77) was rated significantly higher than both the 90° (p < 0.05) and 180° (p < 0.001) positions, suggesting that the visibility of the pull-tab does not always correspond directly to higher taste expectations. The Pineapple condition (M = 7.06, SD = 1.44) was rated significantly higher than all Can conditions (all ps < 0.001), supporting our primary hypothesis.

Figure 3 Overall likings for the six images (0 deg, 90 deg, 180 deg, 270 deg, Down, and Pineapple).

Statistically significant differences are indicated by asterisks (* p < 0.05). Error bars represent standard deviation (SD).

Table 1 Summary of overall liking for six images.

Image	Comparison	Estimate	t-value	95% CI	p-value	
				Lower	Upper		
Pineapple	0 deg	2.190	14.633	1.896	2.485	<0.001***	
	90 deg	2.833	18.927	2.539	3.128	<0.001***	
	180 deg	3.107	20.756	2.813	3.401	<0.001***	
	270 deg	2.726	18.212	2.432	3.020	<0.001***	
	Down	2.679	17.893	2.384	2.973	<0.001***	
0 deg	90 deg	0.643	4.294	0.349	0.937	<0.001***	
	180 deg	0.917	6.124	0.622	1.211	<0.001***	
	270 deg	0.536	3.579	0.242	0.830	<0.001***	
	Down	0.488	3.261	0.194	0.782	0.001**	
90 deg	Down	−0.155	1.034	−0.449	0.139	0.302	
	180 deg	0.274	1.829	−0.020	0.568	0.068	
	270 deg	−0.107	0.716	−0.401	0.187	0.475	
180 deg	270 deg	−0.381	2.545	−0.675	−0.087	0.011*	
	Down	−0.429	2.863	−0.723	−0.134	0.004**	
270 deg	Down	−0.048	0.318	−0.342	0.247	0.751	
Notes.

Post-hoc comparisons of overall liking scores between image conditions.

* p < 0.05.

** p < 0.01.

*** p <0.001.

Discussion

The present study aimed to examine whether the ease of opening, as determined by the orientation of a can’s pull-tab, significantly influences taste expectation evaluations when participants observe images of canned food. The findings supported the primary hypothesis that images of canned food would receive lower taste expectation ratings than those of fresh food (mean difference = 3.14, p < 0.001), highlighting the influence of packaging cues on sensory expectations. Furthermore, the study demonstrated that pull-tab positioning significantly affects participants’ taste evaluations. The 0° position (user-friendly orientation) received the highest ratings (M = 4.87), whereas the 180° position was rated lowest (M = 3.95), suggesting that perceived ease of use contributes to the mental stimulation of consumption. The results for the 0° and 180° positions supported the hypothesis. The findings are consistent with previous research indicating that graspable or action-congruent orientations increase perceived attractiveness and facilitate eating-related mental imagery (Shibuya et al., 2022). Contrary to our expectation, the Down condition—representing cans without visible pull-tabs—did not receive the lowest ratings. One possible explanation is that participants’ perception of manipulating such objects. The relatively favorable evaluations of the Down condition might also reflect perceptual simplicity; a less visually distracting layout could direct attention toward the product itself rather than to functional elements like the pull-tab. Additionally, the brief exposure time may have encouraged intuitive, rapid judgements, wherein visually simple stimuli tend to elicit more positive evaluations due to increased cognitive load.

The disparities in evaluation between images of real food and those in cans may be attributed to sensory and experiential aspects related to the act of eating (Palmer, Gardner & Wickens, 2008). Possible explanations for these differences include: (1) Images of real food may evoke sensory memories related to taste, smell, and texture (Avery et al., 2021; Mojet & Köster, 2005), (2) real food images are more likely to be associated with the act of eating (Versace et al., 2019). In contrast, images of food containers, such as cans, may not elicit the same sensory stimulation as images featuring the food itself. Moreover, the sight and smell of real foods, as well as graphic depictions of appetizing foods, can influence eating behavior by triggering food cravings and shaping food choices under certain conditions (Blechert et al., 2014; Zhang & Spence, 2023). The application of neuromarketing techniques in evaluating food packaging can provide valuable insights into food packaging design (Moya, Garcia-Madariaga & Blasco, 2020). As Spence, Motoki & Petit (2022) have shown, a variety of factors can influence the perceived taste of food.

As noted by Spence, Motoki & Petit (2022), the ease of grasping is suggested to facilitate mental simulation of the act of eating, thereby triggering the recall of the eating process in the viewer’s brain. Numerous studies have demonstrated that when it is easier to envision the act of eating, an image receives more favorable evaluations compared with situations in which this mental simulation is less accessible (Avery et al., 2021; Mojet & Köster, 2005; Versace et al., 2019; Elder & Krishna, 2012). While the study employed a control condition to isolate the effect of pull-tab positioning from other biases, it is important to acknowledge that residual biases may still have influenced the results. For instance, participants’ pre-existing attitudes toward canned food and their expectations regarding the ease of opening cans may not have been fully controlled. Future research should consider more comprehensive measures to account for these biases, such as pre-screening participants for their attitudes toward canned food or employing a broader range of control conditions. Additionally, examining how these biases interact with different types of food packaging could provide further insights into the factors influencing taste evaluations.

It has been suggested that this perceptual ease enhances one’s sense of attractiveness (Shibuya et al., 2022; Simmons, Martin & Barsalou, 2005). Shibuya et al. (2022) noted that viewers tend to prefer food images when the presentation allows for easy grasping with their dominant hand and facilitates bringing the food to the mouth. Intriguingly, such orientation preferences did not extend to images of objects that are difficult to grasp, such as landscapes. Thus, Shibuya et al. (2022) proposed that graspability is a major determinant of human aesthetic taste.

One limitation of this study was the exclusive use of right-handed participants. Handedness might influence how individuals perceive and interact with objects, particularly in tasks involving mental rotation or motor imagery. Right-handed individuals may prefer different object orientations compared with left-handed individuals, which could affect their taste expectations. Future studies should therefore include left-handed participants to determine whether handedness modulates the relationship between visual stimuli and taste evaluations. Another limitation of the present study is that it did not include a manipulation check to confirm whether the change in pull-tab position influenced participants’ perceived graspability or ease of use. As a result, it remains unclear whether the experimental manipulation successfully induced the intended psychological effect. In addition, although the presentation of products was randomized, the repeated-measures design—where each participant rated all products twice—may have introduced potential order or familiarity effects. Future research should consider between-subjects designs or counterbalancing strategies to address this issue. It is also essential to recognize that the sample used in this study—right-handed university students—may not fully represent the broader consumer population. Preferences and perceptions among different demographic groups could vary significantly. Another limitation concerns the volunteer-based sampling approach. Because participants chose to participate voluntarily, they may differ systematically from the general population in ways that could influence the results, such as interest in the study topic or prior experience with canned foods. Therefore, caution is warranted when generalizing these findings to broader consumer populations. Future studies should include a more diverse sample, encompassing a range of ages, handedness, and backgrounds to assess whether the observed effects are consistent across different populations. Additionally, investigating other types of food packaging and varying contexts could provide a more comprehensive understanding of how visual stimuli influence taste expectations.

Our findings have significant implications for food marketing and packaging design. The substantial difference in taste expectations between canned and fresh food images suggests that marketers of canned products face challenges in consumer perception. Furthermore, these findings may have relevance for e-commerce environments, where consumers primarily interact with products through visual representations rather than physical handling. Optimizing visual cues such as pull-tab orientation and food presentation could enhance perceived quality and influence purchasing decisions in online settings. To address this, we propose the following:

(1) Packaging designers should prioritize pull-tab positioning that faces the consumer, potentially enhancing perceived taste quality without altering the product.

(2) Marketing campaigns for canned foods might benefit from emphasizing the convenience and long shelf-life of products, while also showcasing the food’s appearance outside the can.

(3) Retailers could consider display strategies that make pull-tabs more visible to consumers, potentially influencing purchasing decisions.

Several potential confounding variables warrant consideration.

(1) Participants’ familiarity with and preferences for canned food may have influenced their ratings; those with positive experiences with canned pineapple might have rated the Can conditions more favorably.

(2) Conversely, individuals who rarely consume canned foods might have shown a bias towards the Pineapple condition.

Future research should control for these factors by assessing participants’ prior experiences and attitudes towards canned foods. Additionally, cultural differences in canned food consumption could impact results, suggesting the need for cross-cultural studies in this area.

To build upon this study and address its limitations, we propose the following avenues for future research:

(1) Conduct cross-cultural studies to examine how cultural differences in food packaging preferences might influence taste expectations.

(2) Investigate the impact of different types of food packaging (e.g., glass jars, plastic containers) on taste expectations to determine whether the effects observed with cans generalize to other packaging types.

(3) Explore the potential for augmented reality in marketing, allowing consumers to virtually ‘open’ canned products and view the contents.

(4) Conduct longitudinal studies to assess how repeated exposure to different packaging designs might influence taste expectations over time.

(5) Investigate how eco-friendly packaging options might interact with taste expectations, given the growing consumer interest in sustainability

It is crucial to acknowledge that the findings of the present study may be influenced by the specific context and sample used. Bias related to pull-tab position and participants’ negative perceptions of canned food might not fully represent broader consumer preferences. Further research with a more diverse sample and varying types of food packaging is needed to validate these results and assess their applicability to other contexts.

Conclusions

In conclusion, this study examined the effect of visual stimuli on taste expectations, exploring how images of food presented in various forms influence taste evaluations. The findings underscored a significant impact of presentation on taste evaluation, revealing that images of food containers, such as cans, were rated lower compared with images of the actual food itself. This disparity may be attributed to the sensory and experiential aspects associated with the act of eating, where images of the food evoked sensory memories and positive associations related to taste, smell, and texture. Furthermore, the study investigated the influence of ease of opening, guided by the position of the pull-tab on a can, on taste evaluation. Results demonstrated that different pull-tab positions affected the ratings for the same image, suggesting that the ease of opening plays a pivotal role in the mental simulation of the act of eating. These findings align with existing cognitive neuroscience research, emphasizing that the ease of opening facilitates mental simulation of eating, leading to higher evaluations of the image.

Supplemental Information

Supplemental Information 1 Dataset

Additional Information and Declarations

Competing Interests

Author Contributions

Human Ethics

Data Availability

Kenichi Shibuya is an Academic Editor for PeerJ.

Kenichi Shibuya conceived and designed the experiments, performed the experiments, analyzed the data, prepared figures and/or tables, authored or reviewed drafts of the article, and approved the final draft.

Mana Miyamoto conceived and designed the experiments, performed the experiments, prepared figures and/or tables, authored or reviewed drafts of the article, and approved the final draft.

The following information was supplied relating to ethical approvals (i.e., approving body and any reference numbers):

Approval for this study was obtained from the Ethics Committee of Niigata University of Health and Welfare approved this study (approval number: 18914-221101).

The following information was supplied regarding data availability:

The raw data is available in the Supplemental File.

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
