# Peer review of "Exploring the impact of visual stimuli on taste expectations and the role of pull-tab position in food packaging"

_PeerJ, doi:10.7717/peerj.20417_

## Round 0.1 · original submission · Major Revisions

· Academic Editor

Major Revisions

**Language Note:** When you prepare your next revision, please either (i) have a colleague who is proficient in English and familiar with the subject matter review your manuscript, or (ii) contact a professional editing service to review your manuscript. PeerJ can provide language editing services - you can contact us at [email protected] for pricing (be sure to provide your manuscript number and title). – PeerJ Staff

Reviewer 1 ·

Basic reporting

"PEER REVIEWER ASSESSMENTS:

OBJECTIVE - Full research articles: Is there a clear objective that addresses a testable research question(s) (brief or other article types: is there a clear objective)?
Yes - there is a clear objective.

EXECUTION - Are the experiments and analyses performed with technical rigor to allow confidence in the results?
Yes - experiments and analyses were performed appropriately.

STATISTICS - Is the use of statistics in the manuscript appropriate?
Yes - appropriate statistical analyses have been used in the study.

INTERPRETATION - Is the current interpretation/discussion of the results reasonable and not overstated?
Yes - the author's interpretation is reasonable.

OVERALL MANUSCRIPT POTENTIAL - Is the current version of this work technically sound? If not, can revisions be made to make the work technically sound?
Probably - with minor revisions

Experimental design

DESIGN - Is the current approach (including controls and analysis protocols) appropriate for the objective?
Yes - the approach is appropriate.

Validity of the findings

PEER REVIEWER COMMENTS:

GENERAL COMMENTS: Overall impression:
The study is original and relevant, offering useful insights into how visual cues like pull-tab position shape taste expectations.

What works well:
Clear objective, thoughtful design, and appropriate analysis. The use of real product visuals strengthens its practical relevance.

Needs improvement:
The sample size is small and narrow in scope. Broader participant diversity and clearer real-world implications would improve generalizability.

Additional comments

The authors may consider clarifying the practical implications of their findings in the discussion section. Expanding slightly on how these results could inform packaging or marketing decisions would strengthen the paper’s applied relevance. A brief note on the limitations of the sample (e.g., student participants) would also be helpful."

·

Basic reporting

-

Experimental design

-

Validity of the findings

-

Additional comments

The authors of the paper “Exploring the impact of visual stimuli on taste expectation and the role of pull-tab position in food containers” report a study in which they presented images of pineapple either in a can with varying pull-tab positions or of the actual fruit. They found that the taste perceptions of the product were higher for the actual fruit and as hypothesized were dependent on the position of the pull-tab: the product with the 0° positioned pull-tab received the most positive rating among all the canned products which is (according to the authors) resulting from the higher perceived “ease of opening”. Altogether, the manuscript is well-written, the theory is neatly leading to the hypothesis, and the methods applied for testing the hypothesis are adequate. There are some remarks from my side that I will outline below that will hopefully help to improve the paper.

1) There are a lot of repetitions within the paper (e.g., in the results section the pattern of results is at least described twice and in the discussion part the sample limitations and the recommendations regarding packaging are dispersed and doubled across the section – this could by improved by restructuring this part and making sure to leave out unnecessary repetitions). Of course, the paper will then become a bit shorter, but this is not necessarily a bad change.

2) Why did the authors decide to work with time limitations during the stimulus presentation and rating process? Is there any rationale that led to this procedure? Please explain in more detail. In stores, consumers usually have (or take) enough time to decide about their purchases, so there might have been no need to restrict the time. On the other hand, it could be interesting to study how consciously the participants decide about the products by restricting the presentation time to a very short time window and testing if the pull-tab position still makes a difference. The discussion section could gain more depth if he authors could discuss the issue of conscious decision making.

3) Why did the authors refrain from including a manipulation check (e.g., asking participants to rate the graspability of the product or something similar?. If such a question were included, it would be possible to check for mediation, which would bolster the author's reasoning. Please discuss this as either a limitation or a recommendation for future research.

4) The design of the study was a repeated subjects design – this means each participant rated all products (even twice). Could it be possible that this had an effect on the ratings (though the presentation was randomized)?

5) The “down” position might be rated as relatively positive due to the perceptual ease (i.e., fewer details distracting individuals’ attention from the product). Maybe the authors want to include this in their discussion.

Finally, on a more general note, I really liked the fact that the journal provides the data for the study (I had a short look at it just to make sure). I hope that this will become a standard for other journals as well, which would make open science so much easier.

---

## Round 0.2 · Minor Revisions

· Academic Editor

Minor Revisions

Thank you once again for submitting your manuscript to PeerJ. I have now reviewed the revised submission along with your detailed response to the reviewers' comments from the previous round.

I would like to extend my sincere gratitude for your patience throughout this iterative review process and your impressive diligence in addressing the previous concerns. The manuscript has been substantially strengthened by the modifications you have made. The manuscript is now in excellent shape. However, before final acceptance, I request that you address a few very minor points related primarily to clarity and presentation.

Reviewer 1 ·

Basic reporting

Figure Enhancement: Figures 2 and 3 could be enhanced for clarity and visual accessibility (e.g., clearer labels on axis titles).

Grammar & Syntax Review: A final pass for minor language issues would polish the manuscript further (e.g., awkward phrasing in a few paragraphs, such as “mean the overall liking ratings…” should be “mean overall liking ratings”).

Consider adding a brief mention of the potential application of these findings in e-commerce environments where visual packaging is the primary interaction point.

Experimental design

-

Validity of the findings

-

·

Basic reporting

-

Experimental design

-

Validity of the findings

-

Additional comments

The authors responded to all concerns with care, and it is notable that the manuscript has improved through the revisions.

There is only one mistake left that needs to be corrected:
The authors added the following sentence on page 5 (line 46-48): “Moreover, the time-limited presentation may have encouraged more intuitive, rapid judgements, in which visually simple stimuli are more likely to elicit favorable evaluations due to reduced cognitive load.”
– Most likely, they mean “increased” (rather than reduced) cognitive load. Please correct or explain.
I thank the authors for handling my comments well.

Reviewer 3 ·

Basic reporting

NA

Experimental design

NA

Validity of the findings

NA

Additional comments

Below are my comments about the paper entitled: “Exploring the Impact of Visual Stimuli on Taste Expectation and the Role of Pull-Tab Position in Food Containers”.
- For the statistical analysis, it looks fine. On the other hand, the explanation of the Kenward-Roger method is a bit too in-depth, thus, there is no need for all the details.
- I would stick to reporting the SD rather than standard error, across the paper.
- No need to report the t value or the F value, the p-value is enough.
- The volunteer method of sampling will limit generalizability. This needs to be highlighted in the limitation section.
- I am not sure if a small table summarizing the characteristics of the subjects included n the study will be helpful.

---

## Round 0.3 · Minor Revisions

· Academic Editor

Minor Revisions

Thank you for the revisions. However, Figures 2 and 3 remain unclear. Could you please clarify what the labels a, b, and c represent in these figures?

Additionally, the statement regarding post-hoc pairwise comparisons using the Bonferroni correction does not appear to correspond with the data presented in Figure 3.

In Figure 2, the meaning of labels a and b is also unclear—please specify what they denote.

Lastly, the final number of participants and their demographic information, which you referenced in the rebuttal letter, is not included in the Results section. Kindly ensure this information is provided.

---

## Round 0.4 · Minor Revisions

· Academic Editor

Minor Revisions

Thank you for the prompt answer. However, the figures are still not clear. Use connecting lines to suggest a significant difference, when applicable that is a connecting line between the bars. Moreover, the annotation you have put is not accurate: (a < b, and c < a and b; p < 0.05

---

## Round 0.5 · accepted · Accept

· Academic Editor

Accept

Thank you for your patience and prompt responses. The current revision has been assessed, and it is ready for publication.